# The Role of Cisternostomy and Cisternal Drainage in the Treatment of Aneurysmal Subarachnoid Hemorrhage: A Comprehensive Review

**DOI:** 10.3390/brainsci13111580

**Published:** 2023-11-11

**Authors:** Alberto Vandenbulcke, Mahmoud Messerer, Roy T. Daniel, Giulia Cossu

**Affiliations:** Department of Neurosurgery, University Hospital of Lausanne (CHUV), University of Lausanne, 1015 Lausanne, Switzerland

**Keywords:** aneurysmal subarachnoid hemorrhage, vascular neurosurgery, ruptured aneurysm, cisternostomy, cisternal drain, delayed cerebral ischemia, cerebral vasospasm, hydrocephalus

## Abstract

Aneurysmal subarachnoid hemorrhage (aSAH) provokes a cascade reaction that is responsible for early and delayed brain injuries mediated by intracranial hypertension, hydrocephalus, cerebral vasospasm (CV), and delayed cerebral ischemia (DCI), which result in increased morbidity and mortality. During open microsurgical repair, cisternal access is achieved essentially to gain proximal vascular control and aneurysm exposition. Cisternostomy also allows brain relaxation, removal of cisternal clots, and restoration of the CSF dynamics through the communication between the anterior and posterior circulation cisterns and the ventricular system, with the opening of the Membrane of Liliequist and lamina terminalis, respectively. Continuous postoperative CSF drainage through a cisternal drain (CD) is a valuable option for treating acute hydrocephalus and intracranial hypertension. Moreover, it efficiently removes the blood and toxic degradation products, with a potential benefit on CV, DCI, and shunt-dependent hydrocephalus. Finally, the CD is an effective pathway to administer vasoactive, fibrinolytic, and anti-oxidant agents and shows promising results in decreasing CV and DCI rates while minimizing systemic effects. We performed a comprehensive review to establish the adjuvant role of cisternostomy and CD performed in cases of direct surgical repair for ruptured intracranial aneurysms and their role in the prevention and treatment of aSAH complications.

## 1. Introduction

Aneurysmal subarachnoid hemorrhage (aSAH) is responsible for 5% of all stroke events, and despite continuous evolution in diagnosis and management, it remains associated with high morbidity and mortality [1,2]. Significant cognitive impairments and persistent neurological deficits are reported in up to 20% of survivors [3,4], and only 6% to 17% of patients return to their previous occupation [5,6,7]. Subarachnoid hemorrhage starts a cascade reaction responsible for early and delayed brain injuries, such as intracranial hypertension, hydrocephalus, cerebral vasospasm (CV), and delayed cerebral ischemia (DCI). All these events are the main factors associated with increased morbidity and mortality [2,4].

International guidelines recommend a complete and early obliteration of the ruptured aneurysm to avoid rebleeding. The treatment technique is subject to multidisciplinary discussion and secondary to surgeons’ experience [2]. 

The endovascular treatment is progressively gaining popularity and is generally recommended over microsurgical clipping when the two techniques are considered equivalent. However, clipping is generally recommended for younger patients presenting with large intraparenchymal hematoma and/or middle cerebral artery aneurysms [2].

Indeed, surgical clipping is generally associated with a higher rate of complete obliteration and subsequent lower risk of rebleeding [2,8]. The access for microsurgical clipping is anatomically favorable for anterior circulation aneurysms, and it allows evacuation of intraparenchymal hemorrhage, cisternal opening, and lavage. Cisternostomy, defined as the opening of basal cisterns, contributes to per-operative brain relaxation, clot evacuation, and CSF flow restoration; this includes an extensive microsurgical opening of the suprasellar cisterns, the lamina terminalis (LT), and the Membrane of Liliequist (MoL) [9,10,11]. The insertion of a cisternal drainage (CD) in the prepontine cistern then allows continuous postoperative CSF drainage from the anterior and posterior circulation cisterns and the ventricular system, thus enhancing the positive effects of citernostomy [11,12].

Aneurysm occlusion represents only the first step of the complex aSAH management. The amount of blood clots and their toxic degradation products are the principal trigger factors for early and delayed brain injury [13,14,15,16,17,18]. The prevention and management of early and delayed brain injuries may significantly improve the functional outcome. Despite improvements in medical research, no definitive treatment is available at present, and the definitive role of endovascular versus open surgery to prevent early and delayed brain injury is controversial. Clot removal has been proposed to block the cascade reaction responsible for early and delayed brain injury following aSAH [19,20]. Cisternostomy may improve clot washout; however, complete clot removal is nearly impossible, and continuous CSF drainage seems a valuable option to increase clot evacuation [21,22,23,24,25,26].

External ventricular drain (EVD) is the most common technique, and it may resolve acute hydrocephalus despite CSF circulation reduction and inefficient blood product washout [21,22,23]. On the other hand, cisternal and lumbar drainage may favor the natural CSF circulation with an enhanced effect on toxic product evacuation, especially when the cisternal drain is combined with cisternostomy [12,22,23,25,27,28,29]. Moreover, external drains are not only an output route, but they may be used as an injection pathway for intrathecal therapies. Intrathecal, especially intracisternal, administration has been shown to increase local concentration while minimizing systemic adverse effects [30,31,32,33]. Finally, the CSF flow restoration following LT and MoL opening showed decreased shunt dependency [34], and cisternal drainage seems to magnify its positive effect [11]. Microsurgical cisternostomy combined with the placement of a CD to obtain continuous postoperative CSF drainage may, thus, have multiple roles in the prevention of early and delayed brain injury following aSAH. We performed a comprehensive review of the recent literature to establish the adjuvant role of cisternostomy and cisternal drainage in cases of surgical clipping for ruptured intracranial aneurysms. 

## 2. Methodology

We performed a comprehensive review of the literature on the PubMed database up to July 2023. The aim was to identify articles reporting the use of cisternostomy and cisternal drainage in the treatment of aSAH. The articles were identified using a Boolean search with the keywords “subarachnoid hemorrhage” AND “cisternostomy” OR “cisternal drainage”. Two reviewers (A.V. and M.M.) selected the pertinent articles (Figure 1).

Only original articles in English reporting the microsurgical opening of basal cisterns, LT, and MoL associated with cisternal drainage for the treatment of ruptured intracranial aneurysms were considered. Reference lists were checked to identify relevant studies. Case reports were included only if they reported complications linked to cisternostomy and cisternal drainage. Studies reporting cisternostomy alone or preclinical studies were excluded from the outcome assessment but included in the discussion. 

## 3. Delayed Cerebral Ischemia and Cisternal Drainage

### 3.1. Pathophysiology of Delayed Cerebral Ischemia

DCI occurs in up to 30% of aSAH patients and results in poor functional outcomes in half of them despite adequate treatment [2,13,35]. CV is defined as a narrowing of the angiographically visible cerebral arteries that may occur in 30–70% of patients following aSAH [36]. It has been considered for decades the principal cause of DCI. Recent evidence demonstrated that CV in the major cerebral vessels is a contributing factor and not the determinant one [13]. Blood degradation products in the subarachnoid space trigger a molecular cascade that may lead to CV, microcirculatory dysfunction, excitotoxicity, oxidative stress, and inflammatory cascade [13,37,38]. The glymphatic system is responsible for clearing the CSF toxic products [39]. However, lymphatic vessel disruption and glymphatic system dysfunction following aSAH may exacerbate neuroinflammatory response and contribute to DCI [40,41,42]. Continuous CSF drainage seems a valuable option to increase clots evacuation, reduce DCI, and improve functional outcomes, but contrasting results are reported with different output pathways [21,22,23,24,25,26]. The studies reporting the impact of external CSF drains in CV and DCI are summarized in Table 1.

Moreover, increased intracranial pressure combined with micro and macrovascular changes may further reduce cerebral perfusion pressure and have been related to the risk of developing hypoperfusion and ischemia [43,44,45]. DCI is a complex process resulting from multiple pathological pathways secondary to aSAH and has been correlated with the extent of subarachnoid hemorrhage, as expressed by the modified Fisher grade [14,46,47]. The role of vasospasm has probably been overemphasized, and a therapeutic approach to all these factors should be preferred. 

### 3.2. Cisternal Drainage, Cerebral Vasospasm and Delayed Cerebral Ischemia

Extensive and early evacuation of cisternal clots shortly after aSAH and before the process of blood degradation takes place could be a valid option to stop the pathological cascade and prevent DCI. Evacuation of a subarachnoid hemorrhage within 48h showed a reduction in CV and DCI in preclinical studies [48,49]. Moreover, the amount of postoperative clots and clot-clearance rate on serial brain CT seems to predict the risk of CV, consequent infarction, and unfavorable outcomes [50,51].

Yasargil pioneered the interest in understanding the anatomy of the basal cisterns. He was the first to report routine basal cistern opening, including LT (Figure 2) and MoL (Figure 3) fenestration to obtain brain relaxation, proximal exposure, extensive evacuation of the subarachnoid hemorrhage, and CSF flow restoration [52]. In the following years, the advent of microneurosurgery allowed widespread diffusion of cisternal procedures [52,53,54]. 

Extensive subarachnoid clot removal was first described by Suzuki and Yoshimoto in 1976 [55]. Early surgery and clot evacuation within 24 h from the bleeding greatly lessen the risk of CV and improve functional outcomes, especially in patients with high Fisher grades aSAH [55,56]. A few reports published in Japan in the seventies and eighties confirmed the positive impact of early operation and clot evacuation on CV and delayed ischemia [50,57,58,59,60]. Moreover, the Japanese register showed an association between surgical clipping and a lower risk of cerebral infarction [61]. 

A few studies have been published dealing with the impact of cisternostomy on CV, DCI, and functional outcome, and the results are summarized in Table 2 [11,12,25,26,28,29,62,63]. 

To our knowledge, Ito et al. were the first to combine the insertion of cisternal drainage with cisternostomy to enhance their positive effects [12]. Cisternal drainage improves toxic substance clearance compared to EVD and may remove more than 1g of hemoglobin per day [26]. Its position in the basal cisterns creates a gravitational gradient to CSF circulation and clearance similar to lumbar drains but achieving it closer to the bleeding source [21,22,24,27]. Moreover, despite the emerging efficacy of LD in reducing DCI and unfavorable outcomes, obstructive hydrocephalus and compressed basal cisterns remain the main limitations in the use of LD, while both situations are efficiently treated with CD [22,27]. Indeed, LT and MoL fenestration combined with CD increase daily CSF drainage, resolve obstructive hydrocephalus, and were significantly associated with a reduction of symptomatic CV [12]. Kawakami et al. performed a continuous cisternal drainage for a minimum of fourteen days in 22 patients. Moderate CV occurred in 22% of cases, and good neurological outcome was observed in 95% of cases [28]. Inagawa et al. evaluated the efficacy of continuous cisternal drainage on CV for 140 consecutive patients. They classified patients according to the total amount of drainage regardless of the duration. Drainage of cisternal CSF of more than 500 mL was shown to be associated with decreased DCI, angiographic and symptomatic CV, and improved functional outcome [29]. Sakaki et al. reported that early clot evacuation combined with CD was associated with better neurological outcomes (*p* = 0.01). Moreover, they observed a reduction of 50% for symptomatic CV and a less severe angiographic spasm [26]. Ogura et al. assessed the efficacy of continuous postoperative cisternal drainage in a retrospective study and showed that it reduced CV and mortality incidence and improved outcomes in patients with severe clinical and radiological presentation [25]. Moreover, continuous postoperative CSF drainage decreases ICP, is directly related to cerebral perfusion, and contributes to DCI [25,63,64]. Cisternostomy may, thus, represent a valuable adjunct in ruptured aneurysms surgery [19,56,65,66,67], as it may allow clots evacuation and postoperative cisternal drainage [12], thus blocking the cascade responsible for DCI [12,25,26,28,29]. Therefore, cisternal drainage, as part of the surgical treatment, has several advantages that allow ICP control and the reduction of several processes that contribute to the development of DCI.

### 3.3. CD and Drugs Administration

Intrathecal drug administration presents many anatomic and pharmacodynamic advantages compared to systemic administration. A higher drug concentration can be reached with minimal systemic effects [68]. Cisternal injections, compared to ventricular and lumbar administration, allow drug delivery directly in the basal cisterns. It may enhance the pharmacologic effects on the large proximal arteries and improve the intraparenchymal diffusion through the perivascular Virchow spaces. Animal models confirmed a more effective and durable vasoactive response compared to intra-arterial administration [32,33]. Prophylactic injections of intrathecal nicardipine and milrinone have shown a reduction in angiographic CV and DCI and an increase in mean cerebral blood flow but without a significant improvement in functional outcomes [69,70,71,72,73,74]. Intraventricular nicardipine showed a significant reduction in DCI and improvement of functional outcomes in patients treated for significant CV [75]. Shibuya et al. administered 2 mg of cisternal nicardipine three times a day for 10 days in 50 patients treated for an aSAH. Prophylactic nicardipine reduced the incidence of radiographic and symptomatic CV by 26% and 20%, respectively, and increased early good clinical outcomes by 15%. However, no statistical significance was reached [72]. Similar results were obtained by Suzuki et al. [69]. 

Magnesium sulfate showed several potential beneficial effects in aSAH patients, such as vasodilatation and attenuation of neuronal death. However, intravenous administration failed to prevent DCI and improve functional outcomes [76], while continuous intracisternal administration significantly improved DCI and functional outcomes [30]. 

In our institution, we introduced intrathecal nicardipine as a treatment for moderate (>50% arterial narrowing at angioCT) to severe (>75% arterial narrowing at angioCT) CV in patients with aSAH in 2019. In our experience, intrathecal nicardipine showed a significant reduction in DCI rate and improved functional outcome (unpublished data). Moreover, cisternal administration showed a positive trend toward a further reduction of DCI when compared to ventricular administration. 

## 4. Cisternal Drainage and Hydrocephalus

Hydrocephalus is one of the major causes of morbidity following aSAH. Acute hydrocephalus is mainly obstructive, while chronic hydrocephalus is more frequently malabsorptive, but its definitive physiopathology continues to be debated [77]. Treatment varies depending on the perceived physiopathology.

### 4.1. Acute Hydrocephalus

Acute hydrocephalus occurs during the first 3 days after aSAH [78], which is reported in 15% to 87% of patients [79]. The pathogenesis is multifactorial, and two major mechanisms are generally accepted: CSF obstruction and malabsorption. In the first case, there is a blockage of CSF circulation in the ventricular system, generally at the level of the aqueduct of Sylvius or the outlets of the fourth ventricle, while in the latter, an increased resistance to CSF outflow is present.

Increasing age, the amount of intraventricular and subarachnoid hemorrhage, hypertension, and posterior circulation aneurysms are accepted risk factors for acute hydrocephalus [2,80,81]. External ventricular drainage is considered safe and efficient to treat acute hydrocephalus [82,83,84], but intraventricular hematoma or collapsed ventricle may cause EVD obstruction, and the rate of blood product degradation clearance from the basal cisterns is often suboptimal [22,23,24]. Despite promising results in CSF clearance and ICP control, LD is contraindicated in the case of obstructive hydrocephalus [22].

During cisternostomy, basal cisterns are widely dissected, and LT (Figure 2) and MoL (Figure 3) are fenestrated. LT fenestration allows CSF drainage from the supratentorial ventricles with a resolution of acute hydrocephalus. However, diffuse subarachnoid hemorrhage increases basal cistern pressure [51,85] and may be responsible for reduced patency of the LT fenestration [86]. Cisternal drainage lowers cistern pressure to near-atmospheric pressure and may reverse the pressure gradient between parenchyma and basal cistern [87], thus representing a valid alternative to EVD [12,26].

### 4.2. Chronic Hydrocephalus

Shunt-dependent chronic hydrocephalus (SDCH) is seen in up to 30% of patients following aSAH [15,88,89]. SDCH is non-obstructive hydrocephalus: aSAH seems to induce fibrosis of the natural CSF resorption pathway (leptomeninges, arachnoid granulations, and glymphatic space), leading to CSF malabsorption and hydrocephalus [90,91,92,93]. Different risk factors are associated with SDCH, such as increasing age, female sex, acute hydrocephalus, and the amount of blood in the ventricular system and subarachnoid space [94,95,96,97]. SDCH is associated with poor neurological outcomes [81,94,98]. Classic management is ventriculoperitoneal (VP) shunt insertion, though this can be complicated by a significant rate of infection and dysfunction in up to 50% of cases [2,99,100]. SDCH prevention may, therefore, reduce these morbidities and thereby improve clinical outcomes. Based on this physio-pathological assumption, peri-operative blood clot removal and increased postoperative clearance directly from the basal cisterns could reduce perivascular space exposure to toxic blood degradation products and consequently prevent fibrosis and SDCH [41,88,101,102,103,104]. 

Surgical variables, such as LT and MoL fenestration during open microsurgical clipping, may improve CSF circulation, decrease arachnoid fibrosis and vascular inflammation, and, consequently, reduce SDCH [19,34,67,79,105]. LT opening and clot removal are associated with a reduced incidence of shunt dependence in some surgical series [65,67,105,106]; however, a significant difference was not supported by a systematic review [79].

Following aSAH, the MoL is thickened and inflamed, creating CSF loculations, also known as the “fifth ventricle,” with consequent CSF blockage at the interpeduncular and prepontine areas (Figure 3a) [34]. MoL opening may, thus, create communication between infratentorial and supratentorial cisterns and seems to be more efficient than LT fenestration alone [34,105,107] in restoring the CSF flow [52,107]. Winkler et al. showed a drastic reduction in SDCH rate in a retrospective series of 663 patients combining fenestration of LT and MoL [34]. 

The adjuvant use of continuous CSF drainage and clearance of the toxic blood degradation products may prevent subarachnoid fibrosis and SDCH. EVD is usually recommended to treat acute hydrocephalus [2]. However, CSF drainage from the lateral ventricle through an EVD could contribute to blood stasis within the basal cisterns. On the contrary, the combination of cisternostomy (basal cistern opening, LM, and MoL fenestration) with CD seems to improve CSF circulation from the lateral ventricles through the subarachnoid space [12,21,22]. Moreover, the CD has been shown to be an efficient washout system with a high amount of hemoglobin drained [26]. 

The impact of cisternal drainage on SDCH is still controversial (Table 3). We recently reported a series of 89 patients treated for aSAH with EVD or CD in our institution. Patients treated with a CD showed an extremely low rate of SDCH (10%), significantly inferior to the EVD cohort, and that was not related to aneurysm treatment modality (endovascular or surgical treatment) [11]. Kawakami et al. reported a retrospective series of 22 patients treated surgically for aSAH: cisternostomy and CD were associated with a low rate (14%) of SDCH [28]. Yamamoto et al. retrospectively reviewed 205 patients who underwent early clipping for ruptured intracranial aneurysm with and without postoperative cisternal drainage. Patients within the CD group showed significant ventricular enlargement (33% vs. 17% *p* = 0.02) at follow-up, while no difference was observed for VP shunt placement (22% vs. 13% *p* = 0.13). However, patients in the CD groups had worse clinical and radiological presenting conditions [62].

Though several studies have shown the value of CSF drainage, the amount of daily drainage necessary to prevent potential complications needs to be studied further as this could have an impact on the neurological outcome. Ogura et al. report a series of 132 patients operated for an aSAH with and without CD. The cisternal drainage group was associated with a favorable clinical outcome. However, the patients who had drainage superior to 250 mL per day had an increased risk of SDCH [25]. The pathophysiology of this occurrence has been studied in some reports. High CSF volume drainage is secondary to increased resistance to CSF outflow and is associated with a higher risk of shunt dependency, independently of the type of drainage (CD, EVD, or LD) [25,63,108,109,110]. Daily CSF output over 200 mL at normal ranges of draining pressure has been reported as a reliable threshold to predict SDCH [110,111]. Instead, high drainage volume at low draining pressure may induce chronic malabsorption. It is argued to cause a collapse of the subarachnoid space, reducing CSF flow blood removal and aggravating the subarachnoid fibrosis [25]. Moreover, excessive drainage leading to intracranial hypotension could result in reduced absorption and increased CSF production, which may promote SDCH [63]. 

Though cisternostomy and cisternal drainage allow clots to wash out, it improves CSF dynamics, thereby reducing shunt dependency. However, drainage volume and draining pressure should be cautiously monitored to avoid excessive drainage and deleterious effects. Further high-quality studies are necessary to evaluate the definitive role of CD on SDCH. 

## 5. Complications

Despite the improvements in microsurgical techniques, cisternostomy remains a delicate procedure, above all, in the context of aSAH, as the different arachnoid layers may be hidden by a large amount of blood. The internal carotid and anterior communicating complex should be gently manipulated. In particular, the surgeon should be aware of the risk of vascular injury, especially for the small branches, such as the recurrent artery of Heubner that may have variable anatomy, the anterior choroidal artery, and the perforators. However, several series have documented the safety of this procedure with respect to vascular injury [11,12,25,34,65]. Furthermore, the opening of basal cisterns and LT may help in further dissection and in gaining proximal vascular control due to brain relaxation [65]. 

Vascular injuries related to the placement of CD are extremely rare, and only a few case reports exist that document the associated vascular injury [112,113]. 

The cisternal drain has a potential risk of infection: the reported infection rate varies from 3 to 9% in the literature and is similar to other drain-related infections [11,25,26].

## 6. Future Directions

Despite multiple reports suggesting a positive impact of cisternostomy and cisternal drainage in the management of CV, DCI, and SDCH, we have no high-quality evidence from studies comparing cisternostomy and CD positioning with standard care or coiling. Even so, due to the encouraging results from studies showing multiple positive effects of cisternostomy and CD in preventing delayed brain injury and improving functional outcomes [11,12,25,26,28,29,62], it has been recently proposed as an adjuvant treatment after endovascular coiling [30]. However, no data are available for this application. 

Finally, despite preliminary promising results on the cisternal administration of vasoactive agents [32,33,68], a comparative study with ventricular administration, including a large cohort of patients, is required to clarify the pros and cons of this administration pathway. 

Future high-quality studies are mandatory to confirm these findings in an effort to improve the outcomes of aSAH patients.

## 7. Conclusions

Cisternostomy and CD positioning, with continuous cisternal CSF drainage, seems to be a valuable and safe option for vascular surgeons to reduce the burden of complications related to aSAH, as they may help in removing toxic blood products responsible for CV, DCI, and SDCH. 

The improvement of CSF dynamics and the restoration of a drainage pathway could also result in a reduction of SDCH. 

Cisternal administration of vasoactive drugs could be an alternative to systemic administration with fewer systemic side effects. Its superiority in intraventricular administration needs to be clarified. 

Thereby, cisternostomy and CD use might improve the clinical outcomes of aSAH patients, especially for high Fisher grades, and further studies with large cohorts are required to confirm these findings. 

## Figures and Tables

**Figure 1 brainsci-13-01580-f001:**
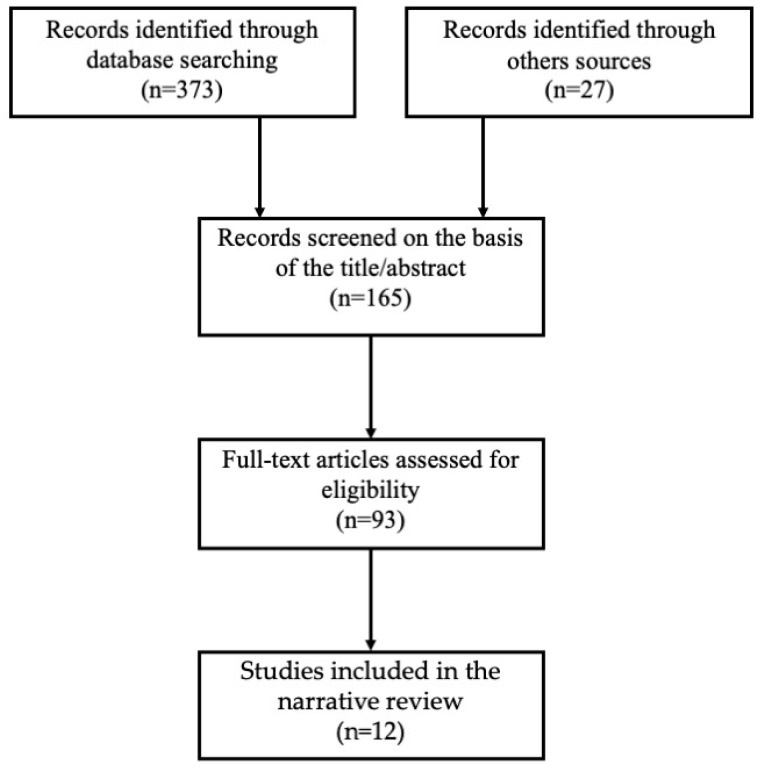
Flow chart showing the search strategy for the narrative review.

**Figure 2 brainsci-13-01580-f002:**
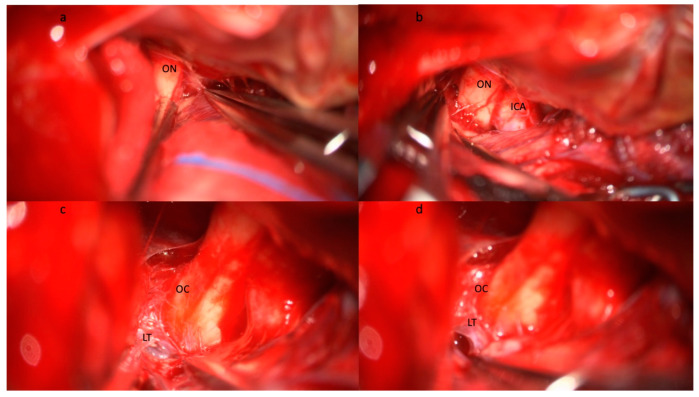
Step-by-step illustration of a cisternostomy performed during microsurgical clipping of an aneurysm of the right middle cerebral artery bifurcation. (**a**) Subfrontal approach and identification of the optic nerve (ON) with (**b**) progressive opening of the optico-carotid cistern and exposure of the internal carotid artery (ICA). (**c**) The dissection continues medially and posteriorly following the ON until the exposure of the optic chiasma (OC) and the lamina terminalis (LT). (**d**) The opening of the lamina terminalis gives access to the third ventricle and allows CSF drainage from the ventricular system. Brain relaxation is generally obtained at this stage after blood clot evacuation from the basal cisterns, the opening of the optico-carotid cisterns, and the LT. Abbreviations: CSF: cerebrospinal fluid; ICA: internal carotid artery; LT: lamina terminalis; OC: optic chiasma; ON: optic nerve.

**Figure 3 brainsci-13-01580-f003:**
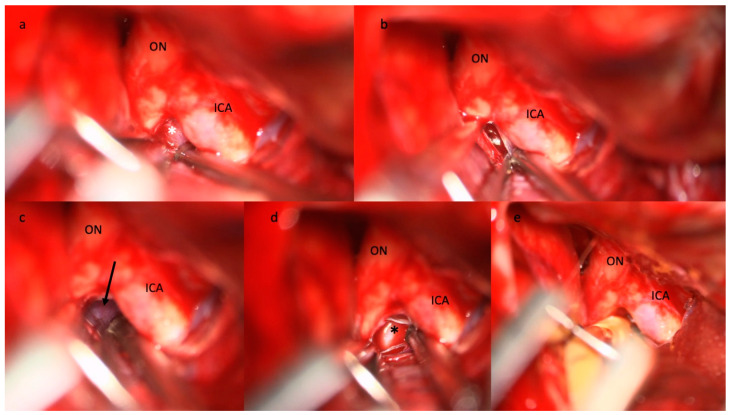
Step-by-step illustration of a cisternostomy performed during microsurgical clipping of an aneurysm of the right middle cerebral artery bifurcation. (**a**) The diencephalic leaf of the Membrane of Liliequist (white asterisk) is seen through the optico-carotid triangle and (**b**) progressively opened, (**c**) enabling the exposure of the mesencephalic leaf (black arrow) that is finally opened. (**d**) The basilar artery (black asterisk) is visible in the pre-pontine cistern. (**e**) The tip of the cisternal drain is positioned between the ICA and the ON in the pre-pontine cistern. Abbreviations: ICA: internal carotid artery; ON: optic nerve.

**Table 1 brainsci-13-01580-t001:** The studies investigating the role of the lumbar drain, external ventricular drain, and cisternal drain for cerebral vasospasm and delayed cerebral ischemia management are summarized here. CD: cisternal drain; DCI: delayed cerebral ischemia; EVD: external ventricular drain; LD: lumbar drain; RCT: randomized controlled trial; STD: standard.

Author and Year	Design	Intervention	Outcome
Wolf et al., 2023 [22]	RCT	LD (144) vs. STD care * (143)	LD reduced DCI (*p* = 0.04) and unfavorable outcomes at 6 months (*p* = 0.04).
Al-Tamimi et al., 2012 [23]	RCT	LD (105) vs. STD care * (105)	LD showed DCI reduction (*p* = 0.021) but no outcome improvement at 6 months.
Maeda et al., 2013 [21]	Retrospective	LD (34) vs. EVD (17)	LD showed more rapid clot washout and a trend toward DCI reduction.
Klimo et al., 2004 [24]	Retrospective	LD (81) vs. STD care * (86)	LD reduced CV (*p* < 0.001) and DCI (*p* = 0.05).
Ogura et al., 1988 [25]	Retrospective	CD (101) vs. no drain (31)	The CD was associated with no significant reduction of CV, although positive trends were seen in patients with high Fisher grades.
Sakaki et al., 1987 [26]	Retrospective	CD (75) vs. STD care * (74)	Early CD significantly reduced DCI (*p* < 0.05) and improved outcomes (*p* < 0.01).

* Standard care groups included patients treated with or without EVD, depending on the presence of acute hydrocephalus.

**Table 2 brainsci-13-01580-t002:** The studies investigating the role of the cisternal drain in CV and DCI management are summarized here. CD: cisternal drain; DCI: delayed cerebral ischemia; STD: standard.

Author and Year	Design	Intervention	Outcomes
Inagawa T et al., 1991 [29]	Retrospective	CD (140)	The total amount of CSF cisternal drainage was correlated with decreased CV and DCI.
Ogura et al., 1988 [25]	Retrospective	CD (101) vs. no drain (31)	The CD was associated with no significant reduction of CV, although positive trends were seen in patients with high Fisher grades.
Sakaki et al., 1987 [26]	Retrospective	CD (75) vs. STD care * (74)	Early CD significantly reduced DCI (*p* < 0.05) and improved outcomes (*p* < 0.01).
Kawakami et al., 1987 [28]	Retrospective	CD (22)	Symptomatic CV occurred in 22% of cases and good functional outcome in 95% of cases.
Ito el al., 1986 [12]	Retrospective	CD (38)	Effective cisternal drainage was correlated with reduced CV and improved outcomes.

* Standard care groups included patients treated with or without EVD, depending on the presence of acute hydrocephalus.

**Table 3 brainsci-13-01580-t003:** The main characteristics of the studies investigating the role of the cisternal drain in shunt-dependent chronic hydrocephalus prevention are reported here. CD: cisternal drain; EVD: external ventricular drain; SDCH: shunt-dependent chronic hydrocephalus.

Author and Year	Design	Intervention	Result
Garvayo et al., 2022 [11]	Retrospective	CD (22) vs. EVD (67)	CD significantly reduced the SDCH rate compared to EVD independently of the treatment modalities (*p* = 0.02).
Kasuya et al., 1991 [63]	Retrospective	Combination of CD, LD and EVD (92) vs. no drain (16)	Reported a significant relationship between high CSF volume drainage at lower heights of drainage and SDCH (*p* < 0.005).
Ogura et al., 1988 [25]	Retrospective	CD (101) vs. no drain (31)	Cisternal drainage <250 mL/die improved functional outcome without significant impact on SDCH. Excessive cisternal drainage (>250 mL/die) significantly increased the risk of SDCH (*p* < 0.05).
Yamamoto et al., 1987 [62]	Retrospective	CD (136) vs. no drain (69)	CD significantly improved functional outcomes without a significant effect on SDCH.
Kawakami et al., 1987 [28]	Retrospective	CD (22)	Reported a low rate of SDCH (14%).

## Data Availability

No supplementary data are available.

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
