# Peer review of "The Role of Cisternostomy and Cisternal Drainage in the Treatment of Aneurysmal Subarachnoid Hemorrhage: A Comprehensive Review"

_brainsci, 2023, doi:10.3390/brainsci13111580_

Round 1

Reviewer 1 Report

Comments and Suggestions for Authors

The authors made a comprehensive review of cisternostomy in subarachnoid hemorrhage.  But the definition of "cisternostomy" seems obscure.   The theme fluctuate from opening the cistern to adding fenestration of the LT and  ML, or additional drainage of the CSF.   

In addition, cisternostomy and fenestration of the LT and ML is an optional procedure during clipping surgery, which potentially accompanies the risk of injury of arteries such as anterior choroidal artery, or reccurent artery of Heubner.   Considering that there are possible publication bias among those studies reporting efficacy of cisternal drainage or extensive removal of cisternal clot, they should refer to the risk of complication.

Comments on the Quality of English Language

There are some spelling error.

"modified Fisher"(l104) should be "modified Fisher group".

Author Response

We agree that a clear definition of cisternostomy was missing. We defined it in the introduction and we differentiated cisternostomy from cisternal drain positioning, as many authors only open the basal cisterns without positioning a cisternal drain. We also verified that no confusion was made within the text.

We added the following text:

“Cisternostomy, defined as the opening of basal cisterns, contributes to per-operative brain relaxation, clot evacuation and CSF flow restoration. This includes an extensive microsurgical opening of the suprasellar cisterns, the lamina terminalis (LT) and the Membrane of Liliequist (MoL) [9–11]. The insertion of a cisternal drainage (CD) in the prepontine cistern then allows continuous post-operative CSF drainage from the anterior and posterior circulation cisterns and from the ventricular system, thus enhancing the positive effects of citernostomy [11,12].”

We also agree with the reviewer that cisternostomy may be associated with a certain risk of arterial injury and we added the paragraph n 4 to discuss the complications as follows:

“Despite the improvements in microsurgical techniques, cisternostomy remains a delicate procedure, above all in the context of aSAH as the different arachnoid layers may be hidden by a large amount of blood. The internal carotid and anterior communicating complex should be gently manipulated. In particular the surgeon should be aware of the risk of vascular injury, especially for the small branches such as the recurrent artery of Heubner that may have a variable anatomy, the anterior choroidal artery and the perforators. However, several series have documented the safety of this procedure with respect to vascular injury [11,12,25,34,65]. Furthermore, the opening of basal cisterns and LT may help in further dissection and in gaining proximal vascular control, due to brain relaxation[65].

Vascular injuries related to the placement of CD are extremely rare and only a few case reports exist that document the associated vascular injury [112,113].

The cisternal drain has a potential risk of infection: the reported infection rate varies from 3 to 9 % in literature and is similar to other drain related infections [11,25,26].”

Reviewer 2 Report

Comments and Suggestions for Authors

Dear Authors,

I am glad to have the opportunity to review your work.  The topic has interest for readers, but some important shortcomings must be addressed.

1.      Please state clearly what the aim of your paper is. You have not stated it in the abstract nor in the main text, after Introduction section.

2.      Please add section Methodology. State what is the study design, how you perfomed literature search and data extraction. Also, what are the inclusion/exlusion criteria

3.      Please add Flow chart which will explain which number of included studies as well as excluded studies and for which reason

4.      I suggest to remove Figures from the Introduction section into different section.

5.      Add Future directions section, before the Conclusion

I suggest revision of the paper.

Author Response

  1. The aim of the study was already stated in the abstract and in the introduction paragraph, we modify it in order to enhance it in the abstract:

“We here performed a comprehensive review to establish the adjuvant role of cisternostomy and CD performed in cases of direct surgical repair for ruptured intracranial aneurysms and their role in the prevention and treatment of aSAH complications.”

And in the introduction as follows:

“We performed a comprehensive review of the recent literature to establish the adjuvant role of cisternostomy and cisternal drainage in cases of surgical clipping for ruptured intracranial aneurysms.”

2/3.     We agree with the reviewer that a systematic review has some methodological advantage and stronger impact. However, the aim of our paper is to provide a wide presentation of cisternostomy and cisternal drainage role in aSAH patients. Indeed, we report an historical overview of all the different application fields (CV, DCI, intrathecal injections, acute and chronic hydrocephalus). For those reasons we estimate that a narrative review provides a more comprehensive overview. We have modified the title to clarify that it is a comprehensive review and not systematic. For all those reasons, no methodological paragraphs or flow chart are possible.

  1. We agree with the reviewer that the introduction is not the most adapted section to put the figures. As suggested, we move to figures from the introduction to the paragraph 2.2 “Cisternal drainage, cerebral vasospasm and delayed cerebral ischemia” .

  1. As proposed with add a paragraph with future directions as follows:

Despite multiples reports suggesting a positive impact of cisternostomy and cisternal drainage in the management of CV, DCI and SDCH, we have no high-quality evidence from studies comparing cisternostomy and CD positioning with standard care or coiling. Even so, due to the encouraging results from studies showing multiple positive effects of cisternostomy and CD in preventing delayed brain injury and improving functional outcomes [11,12,25,26,28,29,62], it has been recently proposed as an adjuvant treatment after endovascular coiling[30]. However, no data are available for this application.

Finally, despite preliminary promising results on cisternal administration of vasoactive agents [32,33,68], a comparative study with ventricular administration including a large cohort of patients is required to clarify the pro and the cons of this administration pathway.

Future high-quality studies are mandatory to confirm these findings in an effort to improve the outcomes of aSAH patients.”

Reviewer 3 Report

Comments and Suggestions for Authors

The authors reported a review of the role of cisternotomy in the treatment of SAH. After careful analysis, I have some suggestions:

- The article is a "Narrative review" and not a Systematic review with PRISMA criteria. If the author would give more scientific impact, they should use PRISMA criteria.

- The authors do not write the aim of the article. It is essential for the reader.

- English needs a revision, some grammar and syntax was wrong.

- A local patient series could improve the interest of the paper. 

-  Any speculation/ hypothesis about the improvement after cisternotomy?

Comments on the Quality of English Language

- English needs a revision, some grammar and syntax was wrong.

Author Response

-We agree with the reviewer that a systematic review has some methodological advantage. However due to the heterogeneity of the studies and aspects discussed in the paper we estimate that a narrative review may be more adapt to the scope of the paper. We did not use PRISMA criteria for the article search but a PICO question to select the pertinent articles for our article.

- The aim of the study is stated in the abstract and in the introduction paragraph as follows:

“We performed a narrative review of the recent literature to establish the adjuvant role of cisternostomy and cisternal drainage in cases of surgical clipping for ruptured intracranial aneurysms”.

-We perform a grammar check as requested and the text was revised by an English-native speaker.

- We agree that a case series may improve the strength of the paper. For this reason, we cited the published data from our institution about cisternostomy and shunt dependant chronic hydrocephalus after aSAH in the 3.2 paragraph as follows:  

We recently reported a series of 89 patients treated for aSAH with EVD or CD. Patients treated with a CD showed an extremely low rate of SDCH (10%), significantly inferior to the EVD cohort, and that was not related to aneurysm treatment modality (endovascular or surgical treatment) [11].”

Moreover, we discuss our unpublished data about cisternal drain and intrathecal nicardipine in the 2.3 paragraph. We modify the paragraph in order to give more information and clarify that the data came from our institution as follows:  In our institution we introduced intrathecal nicardipine as treatment for moderate (>50% arterial narrowing at angioCT) to severe (>75% arterial narrowing at angioCT) CV in patients with aSAH in 2019. In our experience intrathecal nicardipine showed significant reduction in DCI rate and improved functional outcome (unpublished data). Moreover, cisternal administration showed a positive trend toward a reduction of DCI when compared to ventricular administration”.  

The final results of our surgical series are under review in another original article.

- We discussed the impact of cisternostomy and cisternal drain on vasospasm, DCI and shunt dependency while advancing some hypothesis on its efficacy along the different paragraphs.

Paragraph 2.2Moreover, continuous post-operative CSF drainage decreases ICP that is directly related to the cerebral perfusion and contribute to DCI [25,63,64]. Cisternostomy may thus represent a valuable adjunct in ruptured aneurysms surgery [19,56,65–67] as it may allow clots evacuation and post-operative cisternal drainage [12], thus blocking the cascade responsible for DCI [12,25,26,28,29]. Therefore, cisternal drainage, as part of the surgical treatment, has several advantages that allow ICP control and the reduction of several processes that contribute to the development of DCI. “

Paragraph 3.2:

Based on this physio-pathological assumption, peri-operative blood clots removal and increased post-operative clearance directly from the basal cisterns could reduce perivascular space exposure to toxic blood degradation products and consequently prevent fibrosis and SDCH [41,88,101–104].”

A future directions paragraph was also add before the conclusion section as follows:

Despite multiples reports suggesting a positive impact of cisternostomy and cisternal drainage in the management of CV, DCI and SDCH, we have no high-quality evidence from studies comparing cisternostomy and CD positioning with standard care or coiling. Even so, due to the encouraging results from studies showing multiple positive effects of cisternostomy and CD in preventing delayed brain injury and improving functional outcomes [11,12,25,26,28,29,62], it has been recently proposed as an adjuvant treatment after endovascular coiling[30]. However, no data are available for this application.

Finally, despite preliminary promising results on cisternal administration of vasoactive agents [32,33,68], a comparative study with ventricular administration including a large cohort of patients is required to clarify the pro and the cons of this administration pathway.

Future high-quality studies are mandatory to confirm these findings in an effort to improve the outcomes of aSAH patients.”

Round 2

Reviewer 2 Report

Comments and Suggestions for Authors

Dear authors,

The paper looks much better now, afte making improvements. Still, you have not responded to the 2. and 3. remark, and have not added Methodology section or Flow chart. I am well aware this is not systematic, but narrative review. If it was systematic review, PRISMA framework would be needed. Since it is only literature review, Methodology section covers the type of study, explanation on how you performed literature search (which databases, which key words, which type of papers, which languages).. etc. According to that, Flow chart should be made. Please make this corrections.

I suggest revision.

Author Response

As suggested by the reviewer we add methodology paragraph (n3) and a flow chart (figure 1) explaining the selection process, as follow:

We performed a comprehensive review of the literature on PubMed database up to July 2023. The aim was to identify articles reporting the use of cisternostomy and cisternal drainage in the treatment of aSAH. The articles were identified using a Boolean searches with the key words “subarachnoid hemorrhage” AND “cisternostomy” OR “cisternal drainage”. Two reviewers (A.V. and M.M.) selected the pertinent articles (Figure 1).

Only original articles in English reporting microsurgical opening of basal cisterns, LT and MoL associated with cisternal drainage for the treatment of ruptured intracranial aneurysms were considered. Reference lists were checked to identify relevant studies. Case reports were included only if they reported complications linked to cisternostomy and cisternal drainage. Studies reporting cisternostomy alone or pre-clinical studies were excluded for outcome assessment but included in the discussion.”

Reviewer 3 Report

Comments and Suggestions for Authors

good improvements 

Comments on the Quality of English Language

ok, good improvements.

Author Response

We performed an English check as suggested.